# Implication between Genetic Variants from APOA5 and ZPR1 and NAFLD Severity in Patients with Hypertriglyceridemia

**DOI:** 10.3390/nu13020552

**Published:** 2021-02-08

**Authors:** Virginia Esteve-Luque, Ariadna Padró-Miquel, Marta Fanlo-Maresma, Emili Corbella, Xavier Corbella, Xavier Pintó, Beatriz Candás-Estébanez

**Affiliations:** 1Cardiovascular Risk Unit, Internal Medicine Department, Hospital Universitari de Bellvitge, 08907 L’Hospitalet de Llobregat, Spain; vesteve@bellvitgehospital.cat (V.E.-L.); mfanlo@bellvitgehospital.cat (M.F.-M.); emilic@bellvitgehospital.cat (E.C.); xcorbella@bellvitgehospital.cat (X.C.); 2Bellvitge Biomedical Research Institute (IDIBELL), 08908 L’Hospitalet de Llobregat, Spain; apadro@bellvitgehospital.cat (A.P.-M.); bcandas@bellvitgehospital.cat (B.C.-E.); 3Medicine and Translational Research, University of Medicine, Universitat de Barcelona, 08036 Barcelona, Spain; 4Clinical Laboratory, Hospital Universitari de Bellvitge, 08907 L’Hospitalet de Llobregat, Spain; 5CIBEROBN Fisiopatología de la Obesidad y Nutrición, Instituto de Salud Carlos III, 28029 Madrid, Spain; 6School of Medicine, Universitat Internacional de Catalunya, 08017 Barcelona, Spain; 7School of Medicine, Universitat de Barcelona, 08907 Barcelona, Spain; 8Clinical Biochemistry, SCIAS-Hospital de Barcelona, 08034 Barcelona, Spain

**Keywords:** non-alcoholic fatty liver disease (NAFLD), hypertriglyceridemia, SNP, APOA5 variant, ZPR1 variant, Lp(a)

## Abstract

Background: Lipid metabolism disorders, especially hypertriglyceridemia (HTG), are risk factors for non-alcoholic fatty liver disease (NAFLD). However, the association between genetic factors related to HTG and the risk of NAFLD has been scarcely studied. Methods: A total of 185 subjects with moderate HTG were prospectively included. We investigated the association between genetic factors’ (five allelic variants with polygenic hypertriglyceridemia) clinical and biochemical biomarkers with NAFLD severity. The five allelic variants’ related clinical and biochemical data of HTG were studied in all the subjects. NAFLD was assessed by abdominal ultrasound and patients were divided into two groups, one with no or mild NAFLD and another with moderate/severe NAFLD. Results: Patients with moderate/severe NAFLD had higher weight and waist values and a higher prevalence of insulin resistance than patients with no or mild NAFLD. Moderate/severe NAFLD was independently associated with *APOA5* rs3134406 and *ZPR1* rs964184 variants, and also showed a significant inverse relationship with lipoprotein(a) [Lp(a)] concentrations. Conclusions: *APOA5* rs3135506 and *ZPR1* rs964184 variants and lipoprotein(a) are associated with moderate/severe NAFLD. This association was independent of body weight, insulin resistance, and other factors related to NAFLD.

## 1. Introduction

Non-alcoholic fatty liver disease (NAFLD) is currently the most common liver disease in developed countries [1] in parallel with the increase of metabolic syndrome and obesity [2]. Its global prevalence in the general population is variable and has been estimated at 52 million in European countries.

Although NAFLD is relatively benign in early stages [3], it can progress to non-alcoholic steatohepatitis (NASH). Progressive NASH may lead to cirrhosis and its complications [4]. NAFLD has well-established risk factors such as insulin resistance associated with overweight, physical inactivity, and type 2 diabetes mellitus [5]. However, epidemiological, familial, and twin studies have clearly indicated that the risk of the development of NAFLD also has a strong genetic component [6]. Therefore, the appearance and development of NASH is attributed to the interaction of genetic and environmental factors [7].

The hallmark of NAFLD is the accumulation of triglycerides in the cytoplasm of hepatocytes that arises from an imbalance between the intake and consumption of lipids. NAFLD is strongly associated with obesity, insulin resistance, and hyperinsulinemia. In these disorders, the increase of the content of fatty acids (FAs) in the liver is caused by an increased release of FAs from the adipose tissue as well as a high “de novo” lipogenesis. This leads to a greater availability of FAs for triglyceride (TG) synthesis, which are incorporated into very-low-density lipoprotein (VLDL) and this, together with the failure of insulin to suppress VLDL production, predisposes to hypertriglyceridemia (HTG) [8]. On the other hand, the excessive contribution and production of FAs in the liver cannot be compensated by the increase in the excretion of TG in the VLDL particles, which is limited by the availability of ApoB100 and, as a consequence, an excessive deposition of TG within the cytosolic lipid droplets (LD) occurs [9].

Clinical HTG has been related to the presence of hepatic steatosis [10,11] in the same way as disturbances in TG metabolism are related to the development of NAFLD. Polygenic HTG is related to the presence of multiple allelic variants since its new re-definition. The presence of these variants together with environmental factors determines clinical HTG. In the studies performed by Hegele et al., the five variants we studied were the most prevalent in population with polygenic HTG. If the presence of these polymorphisms causes alterations in triglyceride metabolism pathways and these alterations are related to NAFLD, it can be expected that single nucleotide polymorphisms (SNPs) associated with HTG could also be related to NAFLD. However, the association between SNPs and HTG and the risk of NAFLD has been scarcely studied.

An inverse relation between Lipoprotein (a) [Lp(a)] and NAFLD as well as with Lp(a) and TG has been found, although little evidence is available in this respect [12,13].

The aim of this study was to analyze the association between NAFLD and lipid and non-lipid risk factors and the five allelic variants shown to have the greatest effect on plasma TG concentrations and clinical HTG [14] according to genome-wide association studies.

## 2. Materials and Methods

### 2.1. Population

A total of 185 patients, aged 18 to 80 years with primary HTG, from a lipid clinic in a tertiary hospital were prospectively included between January 2019 and August 2020. Ultrasonography (US) was performed in all patients within their routine study.

HTG was defined as a serum TG concentration between 2–10 mmol/L on at least two occasions. Patients with severe systemic or life-threatening disease or with the following secondary causes of HTG were excluded: chronic hepatic disease, stage 4–5 chronic kidney disease or on dialysis, type II diabetes mellitus (DM) with insufficient metabolic control (glycated hemoglobin (HbA1c) > 10%) and hypothyroidism with thyroid-stimulating hormone >8 mU/L, drug-induced HTG, and alcohol overuse, defined as a consumption of more than 40 g (four standard drink units (SDU) for males and more than 20 g (two SDU) a day for females) [15].

### 2.2. Variables Selected

All patients provided information about their age, sex, profession, medication, smoking habit, and alcohol consumption. Cigarette smoking was categorized as “none”, “past”, or “current”, and alcohol consumption was classified according to the number of SDU per week. Diet assessment was performed using a validated and standardized questionnaire of the Spanish Atherosclerosis Society (SEA). This is a questionnaire of adherence to heart-healthy diet carried out by the Spanish Atherosclerosis Society in which 14 items are evaluated and a global score is obtained that defines the degree of adherence to this diet [16]. Physical activity was assessed by self-reporting of hours of physical activity per week with only activities of at least moderate intensity such as fast walking, swimming, cycling, or other aerobic activities were considered. Physical activity carried out during travel or basic daily activities was not taken into account. The questionnaire was performed by a trained physician during the regular visit. Anthropometric measures such as weight, height, and waist were measured during the first visit, and the body mass index (BMI) was calculated as weight/height^2^ (kg/m^2^). Data on hypertension, DM, and hypercholesterolemia were also registered.

Variables from lipid profiles corresponding to the lowest, highest, and the most recent TG results with corresponding treatments were recorded for each patient. Analytical results were collected from the electronic clinical records of the patients. Fasting lipid profiles included plasma concentrations of total cholesterol (TC), low-density lipoprotein cholesterol (LDL-c) (when possible), high-density lipoprotein cholesterol (HDL-c), non-high-density lipoprotein cholesterol (non-HDL-c), TG, apolipoprotein B (ApoB), and apolipoprotein A1 (ApoA1). Serum concentrations of aspartate aminotransferase (AST), alanine aminotransferase (ALT), gamma glutamyltranspeptidase (GGT), insulin, basal glucose, HbA1c, Lp(a), homocysteine, albumin, and high-sensitivity C reactive protein (CPR-hs) were also studied. The homeostasis model of insulin resistance (HOMA-IR) was calculated using fasting values in the following formula: HOMA-IR = (fasting glucose (mmol/L) × insulin (µIU/mL))/22.5 [17]. Insulin resistance was considered when HOMA-IR was >2.

All the biochemical analyses were performed in plasma using a COBAS 711 automated analyzer (Roche Diagnostics^®^). HbA1c was measured in serum using an HA-AutoA 1C 8180 auto-analyzer from Menarini^®^.

### 2.3. Genetic Testing

The allelic variants selected for our study were those that were shown by Genowe-wide association study (GWAS) to have the greatest effect on plasma TG concentrations and on the risk of HTG [18], which were: (c.724C > G) from the *ZPR1* gene, (c.56G > C) from the *APOA5* gene, (c.1337T > C) from the *GCKR* gene, (g.19986711A > G) from the *LPL* gene, (c.107 + 1647T > C) from *BAZ1B*, and (g.125478730A > T) from the *TRIB* gene. A higher prevalence than expected by GWAS was found in these patients when these allelic variants were analysed.

The genotypes of the six variants studied were codified as “0” when the variant was non-present, “1” if one allele was present (heterozygous), and “2” if two alleles were present (homozygous).

DNA was extracted from peripheral blood using an automated DNA purification system (Maxwell^®^ RSC Instruments, Promega, Madison, WI) and stored at −80 °C. The following SNPs were genotyped: *ZPR1* gene rs964184 (NM_003904.4:c.*724C > G), *APOA5* gene rs3135506 (NM_001166598.1:c.56G > C), *GCKR* gene rs1260326 (NM_001486.3:c.1337T > C), *LPL* gene rs12678919 (NC_000008.11:g.19986711A > G), *BAZ1B* gene rs7811265 (NM_032408.3:c.107 + 1647T > C), and *TRIB* gene rs2954029 (NC_000008.11:g.125478730A > T).

Genotyping was carried out using the TaqMan SNP Genotyping Assay (assays ID: C_8907629_10, C_25638153_10, C_2862880_1, C_9639494_10, C_2632556_10, and C_15954645_10) Applied Biosystems^®^, Foster City, CA, USA) in 96-well plates that included positive and negative controls. Real-time polymerase chain reaction (PCR) tests were carried out in the 7500 Real-time PCR System, Applied Biosystems (Thermo Fisher Scientific^®^, Waltham, MA, USA) following standard recommendations. Briefly, 1 µL Assay Mix was mixed with 10 µL Supermix SsoAdvanced (Biorad^®^ Hercules, CA, USA), 2 µL genomic DNA (20 ng/µL), and purified water up to 20 µL. The resulting mixture was heated to 50 °C for 2 min and 95 °C for 10 min in a thermal cycler, followed by 40 cycles of denature at 95 °C for 15 s and anneal/extend at 60 °C for 1 min.

### 2.4. Diagnosis of NAFLD

Abdominal US was used to evaluate fatty liver disease in all subjects based on known standard criteria, including hepatorenal echo contrast, liver brightness, and vascular blurring. The radiologists performing the US examination were blinded to the clinical status of the subjects.

The US results were codified as “0” no fatty liver disease, “1” mild or geographic fatty liver disease, “2” moderate fatty liver disease, and “3” severe liver disease.

The assessment of liver echogenicity by US has a high sensitivity and specificity for the detection or exclusion of moderate to pronounced fatty infiltration [19], but has a low accuracy to effectively differentiate between absent and mild steatosis [20]. According to the US results, the patients were separated into two groups, one group without or with mild NAFLD (non/mild NAFLD patients) and another group with moderate or severe NAFLD (moderate/severe NAFLD).

### 2.5. Statistical Analysis

To calculate the statistical power of this study, it was considered that with 93 patients in each group a power of 71.3% existed to detect differences in the contrast of the null hypothesis H_0_: p1 = p2 by means of a bilateral chi-square test for two independent samples, taking into account that the level of significance was 5% and assuming that the proportion in the reference group (mild/moderate) was 3% and the proportion in the moderate/severe group was 13%, with the percentage expressed being the prevalence of the variant *ZPR1*.

The descriptive analysis of categorical variables across groups was shown as frequency and percentages. Normal quantitative variables were expressed as means and standard deviation (SD) and non-normal quantitative variables were expressed as median and interquartile range. Comparisons among groups of NAFLD severity were made using the chi-square or Fisher exact test for categorical variables, ANOVA for normal quantitative variables, and the Mann–Whitney U-test and Kruskal–Wallis for non-normal quantitative variables. The normality of the variables was evaluated using Q-Q plots. The association between the different variables as predictive factors of NAFLD severity was analysed by logistic regression analysis with the variable NAFLD as a dependent variable and the genetic variant as an independent variable. These variables, which showed statistical significance on the bivariate analysis, were included as confounders. The *p* values ≤ 0.05 were considered statistically significant.

## 3. Results

The baseline characteristics of the study participants are shown in Table 1. No differences were found between the two groups of patients in terms of biological sex and age. Compared with non/mild NAFLD patients, patients with moderate or severe NAFLD presented a higher BMI and waist circumference and a higher prevalence of insulin resistance and elevated ALT. Moreover, moderate/severe NAFLD patients presented lower levels of Lp(a), but no differences in other lipid metabolism and inflammatory parameters were found.

No differences were found in clinical or biochemical parameters in the group of non/mild NAFLD patients.

Among the genetic variants studied, the *ZPR1* rs964184 and *APOA5* rs3135506 variants were found to be associated with NAFLD (Table 2).

Logistic regression analysis was used to test the independent association of genetic factors with NAFLD. A recessive model of inheritance best explained the association of NAFLD with the *ZPR1* rs964184 variant (odds ratio (OR) = 4.1). This association was stronger after adjustment for age, gender, and BMI (OR *=* 4.99).

For the *APOA5* rs3135506 variant a real estimation of the association with NAFLD could not be provided because only five homozygous patients were identified. A Fisher exact test was, therefore, made to calculate the relationship with NAFLD, achieving borderline significance (*p* = 0.06) in the recessive model.

When the relationship of at least one of these two variants in homozygosis with NAFLD severity was analysed, a significant association was observed with an OR = 5.44 in the adjusted model.

The results of the logistic regression analysis performed to investigate the effect of genetic and non-genetic factors including Lp(a) serum levels on NAFLD severity are shown in Table 3. Two models were evaluated, one adjusted by BMI and the other adjusted by HOMA-IR as these variables are clinically related. For the model adjusted by BMI the presence of one or both genetic variants in homozygosis showed the strongest association with the severity of NAFLD, as the presence of one or both variants in homozygosis increased the risk of presenting moderate/severe NAFLD (OR = 4.53; CI.18 to 17.41) (*p* = 0.028). Lp(a) showed a significant inverse relation with NAFLD severity (OR = 0.997 95% IC 0.99 to 1) (*p* = 0.048). The BMI was also related to NAFLD, with a variation of one unit of BMI increasing the probability of presenting moderate-severe NAFLD by 10%. For the model adjusted by HOMA-IR the results were similar except for Lp(a). HOMA-IR was also related to NAFLD, with a variation of one unit of HOMA-IR increasing the probability of presenting moderate-severe NAFLD by 17%.

Analysis of the relationship between genetic factors and metabolism was also performed as it is shown in Table 4. No differences in clinical or anthropometric indices were found across carriers or non-carriers of the *ZPR1* (rs964184) or *APOA5* (rs3135506) variants. In terms of metabolic parameters, only differences in TG values were found for *APOA5* (rs3135506) and TC, TG, non-HDL-c, and HDL-c for *ZPR1* (rs964184), with carriers presenting higher concentrations of all these values than non-carriers. No differences were observed in glucose metabolism parameters or inflammatory and hepatic parameters. 

Analysis of the relationship between genetic factors and metabolism was also performed for the rest of the allelic variants studied included in Appendix A. 

## 4. Discussion

NAFLD is a complex metabolic disorder related to alterations in TG metabolism [21]. Although several genes and genetic variants have been identified as being involved in the development of NAFLD [22], the role of TG polymorphisms is not well known.

This was a cross-sectional study of patients with HTG aimed at assessing the clinical, biochemical, and genetic factors related to moderate/severe NAFLD. As expected, the BMI, waist circumference, and insulin resistance were found to be related to NAFLD in this cohort, similar to the results of previous studies [23,24,25]. However, unlike other studies [26], TG levels were not significantly higher in patients with moderate/severe NAFLD compared with those with non/mild NAFLD, which may be because all the patients included in the study had HTG.

The patients studied showed a higher frequency of risk alleles related to HTG than expected by Global Lipids Genetic Consortium (GLGC) studies [14] and this was especially remarkable in relation to the *ZPR1* (rs964184) and *APOA5* (rs3135506) variants. Thus, 8.1% of patients were homozygous for the *ZPR1* variant, whereas the expected frequency according to GLGC studies is about 1%. In addition, 2.7% of patients were homozygous for *APOA5*, while the expected frequency is less than 1% in GLGC studies. The frequencies in which the rest of the variants have been presented are shown in Appendix A. 

Among genetic factors, the *APOA5* rs3135506 variant was found to be related to moderate/severe NAFLD. This could be explained in that this variant has been related to a decrease in hepatic ApoA5 secretion [27] and high hepatic ApoA5 concentrations have been observed in liver biopsies of NAFLD patients and in animal models [28,29].

According to biochemistry studies, the relationship between ApoA5 and NAFLD is not due to the role of Apo5 in stimulating lipoprotein lipase activity at an extracellular level [30] but rather its activity at an intracellular level. ApoA5 affects the number and size of hepatic lipid droplets (LD), whereby high levels of this apolipoprotein lead to an increase in the number and size of LD [31,32]. It has been speculated that ApoA5 plays a role in regulating the directionality of intracellular TG flux [31]. Mature ApoA5 may interact with membrane defects caused by nascent LD formation, leading to its association with nascent LD in the cytosol, promoting the permanence of LD in hepatocytes. It may also pass from the endothelial reticulum lumen to the Golgi and secrete from the cell [33,34].

As far as we know, this is the first study in which a relationship has been found between the *APOA5* rs3135506 variant and moderate/severe NAFLD.

In the present study, a relation between *ZPR1* (rs964184) and NAFLD was also found for the first time. *ZPR1* (rs964184) SNPs correspond to an intergenic zone located near the *APOA5-A4-C3-A1* gene cluster, which has been related to TG concentrations [35,36], and it has been demonstrated that genetic variants in APOA5/A4/C3/A1 gene cluster play an important role in the regulation of plasma triglyceride levels by increased ApoA5 concentration [37]. The variation present in the *APOA5* gene included in this cluster may be directly related to the development of NAFLD; however, since other genes are also affected, functionality studies are needed to explain this association.

It was of note that the presence of both polymorphisms of *APOA5* and *ZPR1* in homozygosity showed a stronger relationship with moderate/severe NAFLD than the presence of other well-known factors, such as obesity or high glucose or TG concentrations.

It has already been demonstrated [38] that the *ZPR1* rs964184 and *APOA5* rs3135506 variants are related to TG concentrations not only in the general population but also in a population with HTG, as in the present study.

To date, few studies have reported an association between Lp(a) and NAFLD. In accordance with the study by Yang et al. [39], Lp(a) concentrations in the present study were lower in patients with moderate/severe NAFLD than in those with non/mild NAFLD. In addition, in a cross-sectional study including 2242 subjects in whom abdominal US was performed and patients were classified according to NAFLD severity and Lp(a) concentrations, Sun Nam et al. observed that Lp(a) concentrations were inversely associated with the presence of NAFLD, but this relation was attenuated after adjusting for insulin resistance [40]. Along the same line, Jung et al. observed that subjects with low Lp(a) and high insulin resistance showed a higher risk for NAFLD than those with high Lp(a) and low insulin resistance, suggesting the opposite association of Lp(a) and insulin resistance [12]. Nonetheless, the mechanism underlying the inverse relationship between Lp(a) and NAFLD is not completely understood.

This study has several limitations. Although the sample size of the study was limited, the patients were selected under strict uniform criteria and the data collection was carried out with a high-quality standard, exclusively by physicians with clinical experience in vascular risk and lipid metabolism disorders, thereby providing strength to the study.

Although liver biopsy is the gold standard for NAFLD diagnosis, it is an invasive and expensive test that is unsuitable for regular screening. The advantages of ultrasound include safety, wide availability, and little associated patient discomfort [19,41] and costs compared with liver biopsy, computerized tomography (CT), and magnetic resonance imaging (MRI), which are also considered as diagnostic tests for NAFLD.

The assessment of liver echogenicity by US has a high sensitivity and specificity for detection or exclusion of moderate to pronounced fatty infiltration [18]. The sensitivity and specificity of ultrasound to detect moderate to severe steatosis using histology as a reference standard are 80–89% and 87–90%, respectively [42,43]. However, abdominal US has low accuracy to effectively differentiate between absent and mild steatosis [20], and the sensitivity and specificity drop to 65% and 81%, respectively, when all grades of steatosis are considered.

## 5. Conclusions

Patients with moderate/severe NAFLD had a higher BMI and waist circumference and a higher prevalence of insulin resistance than patients with mild or without NAFLD. However, only *APOA5* rs3135506 and *ZPR1* rs964184 variants and Lp(a) serum levels were independently associated with moderate/severe NAFLD. Lp(a) showed a significant inverse relation with moderate/severe NAFLD.

## Figures and Tables

**Table 1 nutrients-13-00552-t001:** Clinical and metabolic characteristics of the study subjects.

Variables (*n*)	Non/Mild NAFLD	Mod/Severe NAFLD	*p*
Biological sex (male)	62 (69.7%)	74 (77.1%)	0.253
Age, years	53.3 (10.7)	52.1 (9.8))	0.438
BMI, kg/m^2^	28.1 (3.8)	29.6 (3.9)	0.012
Waist circumference, cm (136)	98.3 (13.4)	104.3 (12.3)	0.008
Alcohol consumption, units/w (174) **	0.0 (0.0–1)		0.697
Diet, SEA (144)	7.1 (3.7)	8.0 (3.4)	0.151
Physical activity, h/week (153) **	1.0 (0.0–3.8)	2.0 (0.0–4.0)	0.580
Total cholesterol, mmol/L (183)	6.78 (1.9)	6.72 (1.8)	0.856
HDL-c, mmol/L (183)	1.00 (0.3)	0.92 (0.3)	0.092
LDL-c, mmol/L (66)	3.48 (1.5)	3.73 (1.7)	0.532
Non-HDL-c, mmol/L (182)	5.77 (1.8)	5.77 (1.9)	0.984
TG, mmol/L (183) **	5.1(4.0–9.6)	6.2(3.5–10.8)	0.439
ApoB, mg/L (179)	1.13 (0.3)	1.20 (0.4)	0.253
ApoA1, mg/L (178)	1.39 (0.2)	1.36 (0.2)	0.400
DM	15 (16.9%)	22 (22.9%)	0.303
Glucose, mmol/L **	5.4 (5.0–6.3)	6.15 (3.5–10.8)	0.356
HbA1c, % (182) **	5.8 (5.4–6.0)	5.9 (5.5–6.3)	0.100
Insulin, pmol/L (133) **	99.1 (71.4–149.8)	126 (82.9–182.0)	0.044
HOMA-IR (131) **	3.6 (2.5–5.7)	4.4 (3.1–7.1)	0.039
Insulin resistance (131) *	51 (83.6%)	66 (94.3%)	0.048
AST, ukat/L **	0.4 (0.3–0.5)	0.5 (0.4–0.6)	0.087
ALT, ukat/L **	0.5 (0.3–0.6)	0.6 (0.4–0.8)	0.040
Albumin (mg/dL) (139)	47.30 (2.6)	47.48 (3.0)	0.706
Lp(a), nmol/L (174) **	54.8 (10–4−184.8)	29 (7.0–98.4)	0.027
Homocysteine, umol/L (109)	10.80 (3.9)	15.59 (5.5)	0.060
CRP-hs, mg/L (119) **	1.0 (0.6–2.0)	1.0 (0.6–3.1)	0.507

Data are expressed as percentage (analysed by chi-square or Fisher test) or mean (+/–SD) for normal quantitative variables (analysed by ANOVA test) or median and interquartile interval for non-normal quantitative variables (analysed by Mann–Whitney U-test). The lipid profile corresponding to the highest TG levels was selected. The highest AST and ALT levels were selected. * Insulin resistance was considered with HOMA-IR >2. Only *n* values other than 185 are described. ** Non-normal quantitative variables. Abbreviations: BMI: body mass index, LDL-c: low-density lipoprotein cholesterol, HDL-c: high-density lipoprotein cholesterol, non-HDL-c: non-high density lipoprotein cholesterol, TG: triglyceride, ApoB: apolipoprotein B, ApoA1: apolipoprotein A1, AST: aspartate aminotransferase, ALT: alanine aminotransferase, HbA1c: glycosylated hemoglobin, Lp(a): Lipoprotein (a), CRP-hs: high-sensitivity C reactive protein, HOMA-IR: homeostasis model of insulin resistance, Diet SEA: standardized diet questionnaire of the Spanish Atherosclerosis Society.

**Table 2 nutrients-13-00552-t002:** Genotype frequencies and odds ratios of variants associated with non-alcoholic fatty liver disease.

		Non/Mild	Moderate/Severe	Genotype Model					
Gene	SNP	*N*	*N*	X2	*p*-Value	OR (95%CI)	Unadjusted *p*-Value	OR (95%CI)	Adjusted *p*-Value
ZPR1	Rs964184			5.27	0.072				
	CC	45 (50.6)	46 (47.9)						
	CG	41 (46.1)	38 (39.6)			0.91 (0.50–1.66)	0.750	0.86 (0.46–1.60)	0.625
	GG	3 (3.4)	12 (12.5)			3.91 (1.04–14.8)	0.044	4.65 (1.20–18.01)	0.026
Dominant model	CG + GG	44 (49.4)	50 (52.1)	0.13	0.719	1.11 (0.62–1.98)	0.719	1.10 (0.61–1.98)	0.765
Recessive model	GG	3 (3.4)	12 (12.5)	5.17	0.023	4.10 (1.12–15.03)	0.034	4.99 (1.33–18.76)	0.017
APOA5	Rs3135506			NA					
	CC	68 (77.3)	66 (68.8)						
	CG	20 (22.7)	25 (26.0)						
	GG	0 (0)	5 (5.2)						
Dominant model	CG + GG	20 (22.7)	30 (31.3)	1.69	0.184				
Recessive model	GG	0 (0)	5(5.2)	NA	0.06 *				
APOA5 + ZPR1				6.05	0.014				
	Non-hom	86 (96.6)	83 (86.5)						
	1 or 2 in hom	3 (3.4)	13 (13.5)			4.54 (1.25–16.53)	0.022	5.44 (1.46–20.25)	0.012

Data are expressed as absolute numbers and percentages by NAFLD severity group (analysed by chi-square or Fisher test). Models of logistic regression analysis unadjusted and adjusted for gender, age, BMI, and genotypes (that were considered as dominant or recessive). * Due to the low frequency of presentation, the relation between the APOA5 variant and NAFLD was analysed with the Fisher exact test. In the dominant model homozygotes and heterozygotes are compared with non-mutant (GG + GC vs. CC), and in the recessive model homozygotes were compared with heterozygotes and non-mutant (GG vs. GC + CC). Abbreviations: OR: odds ratio; CI: confidence interval; SNP: single nucleotide polymorphism; Non-hom: none of the variants are present in homozygous; 1 or 2 in hom: one or both variants are present in homozygosis; NA: not applicable.

**Table 3 nutrients-13-00552-t003:** Factors associated with non-alcoholic fatty liver disease.

Model	Variables	OR	CI	*p* Value Adjusted
Adjusted by BMI	Age	0.983	0.95–1.01	0.297
	Biological Sex (male)	1.232	0.58–2.58	0.582
	Lp(a)	0.997	0.99–1	0.048
	BMI	1.103	1.01–1.20	0.027
	ZPR1 + APOA5	4.537	1.18–17.41	0.028
Adjusted by HOMA-IR	Age	0.963	0.92–1.01	0.081
	Biological Sex (male)	1.513	0.61–3.74	0.582
	Lp(a)	0.998	0.99–1	0.431
	HOMA-IR	1.172	1.03–1.34	0.017
	ZPR1 + APOA5	10.798	1.22–94.89	0.032

Logistic regression analysis was used to test the association of genetic (independent variable), clinical, and biochemical factors (as confounders) with NAFLD severity (dependent variable). Lp(a): lipoprotein a; OR: odds ratio; CI: confidence interval; BMI: body mass index; HOMA-IR: homeostasis model of insulin resistance.

**Table 4 nutrients-13-00552-t004:** Relationship between the genotype and metabolic traits of the *ZPR1* (rs 964184) and *APOA5* (rs 3135506) genes.

ZPR1 (rs 964184)	Wild Type CC	Heterozygous CG	Homozygous GG	*p*
Biological sex (male)	66 (72.5%)	61 (77.2%)	9 (60.0%)	*NA*
BMI kg/m^2^	28.60 (4.4)	29.20 (3.5)	28.40 (2.8)	*0*.*558*
TC mmol/L	6.44 (1.7)	6.82 (1.7)	7.55 (2.1)	0.009
TG mmol/L *	4.55 (3.0–7.9)	6.30 (4.3–11.6)	12.70 (9.4–20.6)	<0.0001
Non-HDL-c mmol/L	5.47 (1.7)	5.80 (1.7)	6.73 (2.18)	0.004
HDL-c mmol/L	0.97 (0.3)	1.00 (0.3)	0.82 (0.2)	0.033
Glucose mmol/L *	5.50 (5.1–6.4)	5.60 (5.1–6.2)	5.30 (4.8–6.8)	0.855
Insulin pmol/L*	95.6 (71.4–134.0)	129 (79.6–176.0)	131 (81.9–191.0)	0.114
HOMA-IR *	3.59 (2.8–5.3)	4.83 (2.6–6.4)	4.23 (2.9–6.9)	0.279
AST µkat/L *	0.45 (0.4–0.6)	0.42 (0.3–0.5)	0.35 (0.3–0.5)	0.125
ALT µkat/L *	0.53 (0.4–0.8)	0.45 (0.4–0.7)	0.42 (0.3–0.6)	0.232
CRP—hs mg/L *	1.00 (0.6–2.8)	0.9 (0.6–2.4)	1.95 (0.5–3.9)	0.765
Lp(a) nmol/L *	37 (7.4–121.8)	47 (8.3–183.4)	18.4 (7–64.8)	0.235
**APOA5** **(rs 3135506)**	**Wild Type** **CC**	**Heterozygous** **CG**	**Homozygous** **GG**	***p***
Biological sex (male)	100 (74.6%)	34 (75.6%)	2 (40.0%)	*NA*
BMI kg/m^2^	28.60 (4.1)	29.71 (3.2)	29.20 (2.8)	0.252
TC mmol/L	6.61 (1.82)	6.83 (1.69)	7.77 (1.67)	0.113
TG mmol/L *	5.18 (3.6–9.5)	7.07 (4.2–12.5)	18.4 (10.6–25.3)	0.002
Non-HDL mmol/L	5.63 (1.83)	5.82 (1.71)	7.03 (1.71)	0.051
HDL-c mmol/L	0.98 (0.33)	0.97 (0.30)	0.73 (0.13)	0.048
Glucose mmol/L *	5.5 (5.1–6.4)	5.6 (5.2–6.2)	5.2 (4.7–6.8)	0.804
Insulin pmol/L *	99.3 (73.4–152.0)	129.00 (88.9–173.5)	138 (98.2–179.5)	0.127
HOMA-IR *	3.63 (2.6–56)	5.28 (3.2–6.5)	4.28 (3.4–4.7)	0.148
AST µkat/L *	0.43 (0.4–0.6)	0.40 (0.3–0.5)	0.38 (0.3–0.5)	0.148
ALT µkat/L *	0.53 (0.4–0.8)	0.42 (0.4–0.8)	0.40 (0.3–0.7)	0.529
CRP-hs mg/L *	0.90 (0.6–2.5)	1.3 (0.5–3.2)	2.05 (0.8–4.0)	0.749
Lp(a) nmol/L *	31.9 (7.4–111.5)	48 (7–190)	50 (16.5–205.8)	0.638

Data are expressed as percentage (analysed by chi-square) or mean (+/−SD) for normal quantitative variables (analysed by ANOVA test) or median and interquartile interval for non-normal quantitative variables (analysed by Kruskall–Wallis test). BMI (body mass index), TC (total cholesterol concentration), Tg (triglycerides concentration), non-HDL-c (total cholesterol except HDL), HOMA-IR (Homeostasis Model Assessment of Insulin Resistance), AST (aspartate amino transferase), ALT (alanine amino transferase), CRP-hs (C-reactive protein high sensitivity), Lp(a) lipoprotein (a), *p*-value of Kruskall–Wallis test. * Non-normal quantitative variables.

## Data Availability

The data presented in this study are available on request from the corresponding author. The data are not publicly available due to respect the privacy of individuals.

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
