# Peer review of "Implication between Genetic Variants from APOA5 and ZPR1 and NAFLD Severity in Patients with Hypertriglyceridemia"

_nutrients, 2021, doi:10.3390/nu13020552_

Round 1

Reviewer 1 Report

General comment - in this cross-sectional observational study, Authors attempt at identifying some possible gene polymorphism, linking hypertriglyceridemia to NAFLD. The topic is new and relevant. The working hypothesis is intriguing. Methods are generally appropriate, except for statistical analysis, which needs in my opinion some refinement. As a consequence, Results show some flaw. In particular, my main concern is the absence of differential analysis for patients with/without metabolic syndrome and/or insulin resistance. Indeed, as Authors correctly state in the Introduction, insulin resistance and metabolic syndrome are well established risk factors for both NAFLD and hypertriglyceridemia. As a personal curiosity, I would like to ask the Authors if they considered the possibility of testing the gene LIPA, coding for lysosomial acid lipase, causing mixed hyperlipemia and liver cirrhosis in the congenital severe deficiency syndrome (Wolman's syndrome) and NAFLD in the mild adult-onset forms (CESD) - for references see also: Baratta F et al, Liver Int, 2019 and Ministrini et al, J Clin Med, 2019).

I cannot recommend the paper for publication in the present form, but I will be glad to consider for publication if the comments below will be appropriately addressed.

Title and abstract - Authors should celarly state in the title that the relationship between gene polymorphisms and NAFLD severity was found in subjects with hypertrigliceridemia (and not in general population or in patients with NAFLD).

Introduction

  1. the research is based on the assumption that a pathophysiological link exists between NAFLD and hypertriglyceridemia, over and beyond the effect of metabolic syndrome itself. This hypothesis is conceivable, but it should be explicitly stated in the introduction and some evidence provided.
  2. Page 1, Line 45 "acquisition and elimination" are quite improper terms, considering the physiology of lipid metabolism in hepatocytes. I suggest using terms like "intake and consumption".
  3. Page 2, Lines 47-48 "In these disorders, the release of fatty acids (FAs) from adipose tissue that are delivered to the liver and de novo lipogenesis are increased" Please, check grammar and syntax in this sentence.

Methods

  1. Authors should describe in the statistical analysis the estimate of sample size, the expected study power (type 2 error probability) and whether 158 is a sufficient number for the given type 1 and type 2 error probabilities. The loss of significance of the Fischer exact test for ApoA5 is due to the low prevalence of this allelic variant (as stated by the Authors). This drives me to suppose that the study is underpowerd.
  2. Authors should also report if normality of continuous variables was tested. All variables were treated as normally distributed, but some of them are usually non-normal (cholesterol, HOMA-IR index, HbA1c). Using parametric tests for non-normally distributed variables may conceal some significant results. I suggest the Authors to check for normality of distributions, if this was not done, and to use appropriate tests.
  3. Page 2, Lines 76-77 "type II diabetes mellitus (DM) and hypothyroidism with insufficient metabolic control (glycated haemoglobin [HbA1c] > 10% and thyroid stimulating hormone > 8 mU/L, respectively)" these are actually two different exclusion criteria.
  4. Page 3, Lines 106-110 - this paragraph needs a reference to the studies that allowed the Authors to identify the 5 alleles.
  5. Page 2, Lines 84-86 "Diet assessment was performed using a validated and standardised questionnaire (SEA) (13), and physical activity was assessed by self-reporting of hours of physical activity per week" Some more details should be provided in this regard: what is the dietary questionnaire investigating? What is the extended definition for SEA? What questionnaire was employed to assess the physical activity level? Furthermore, when expressing physical activity in hours (not METs), the intensity and type of activity should be also reported (i.e. n hours/week of any activity, sports, activity, intense activity, moderate activity etc...)

Results

  1. Authors report the data of prevalence for 2 of the studied alleles only (the significant ones). Data of the remaining alleles should be also reported, even if not significant, at least as supplementary table
  2. Why wasn't ApoA5 discarded, even though the results of the Fisher exact test were non-significant?
  3. Page 6, Lines 208-215. These results are relevant and they should be reported in a Table or Figure in the main text.

Tables

  1. Table 1 - the term "gender" refers to the social and cultural implications of sexual phenotype and to individual self-identification in these socio-cultural structures. If the Authors are referring to biological sex (defined by phenotype), they should use the term "biological sex".
  2. Table 1 - the operation of mixing together patients with ApoA5 and ZPR1 homozygosis is arguable in terms of pure statistics. Indeed, the addition of a neutral factor (ApoA5) to a significant factor (ZPR1) may lead to an apparent improvement of statistical significance because of a sample size effect. This operation could be justified if biologically sounded (e.g. the genes code for proteins in the same pathway).
  3. Table 3 - why wasn't insulin resistance introduced in the model? The difference between the groups reported in Table 1 is significant. It is a potentially relevant confounding factor.

Discussion

  1. Page 6, Line 220. As it is reported in the present paper, the study has a cross.sectional design (not prospective).
  2. Limits of ultrasonographic assessment of steatosis grade (compared to other imaging techniques such as RM or CT), in particular for the quantification of fat content and fibrosis.

Reviewer 2 Report

Overall, the manuscript is interesting because of the novelty of assessment of circulating levels of lipoprotein (a). However, the small sample size and the statistical analyses performed did not completely support the conclusions of the Authors. In addition, the rationale is not well presented, several other polymorphysms are missing and tables are confusing.

Reviewer 3 Report

Here the authors describe an observational study on the possible role of specific genetic variants of Apoa5 and ZPR1 genes in non-alcoholic fatty liver disease (NAFLD) onset and progression. They recruited 185 patients with primary hypertriglyceridemia, which were divided into two groups: non/mild NAFLD and moderate/severe NAFLD. The obtained data showed that two particular variants of the genes Apoa5 and ZPR1, when present in homozygous form could be strictly related to moderate/severe NAFLD.

In my opinion, the study here presented by the authors is well conducted. The abstract proceeds in a logical manner, the introduction is very clear and the results are well described. Lastly, the discussion of the work is well-argued, but as the authors themselves pointed out, there are some criticism and weaknesses in the work.

Despite this, I believe that the work deserves to be published and I suggest the correction of some errors and the insert of some missing parts in the text. I list them below.

  • In lines 25 and 26, the word lipoprotein(a) appears in two different ways, please uniform them.
  • From line 63 to 66, there are two different sizes of the font, please uniform them.
  • In the materials and methods section, in the description of RT-PCR it is very important showing the sequences of the primers used and indicating the genes used as housekeeping, more than one is required, as the guidelines to follow to perform a RT-PCR analysis recommend (https://doi.org/10.1373/clinchem.2008.112797).
  • The caption of table 1 is written with two different font sizes, please uniform them.
  • In line 236, a space between the two numbers of the references in the bracket is required.
  • In line 239, a space at the end of the sentence ending with the word “level” is needed.
  • In line 242, reference 29 is written in italic, please correct it.
  • In line 262, space after the reference number 35 is needed.

Round 2

Reviewer 1 Report

My comments have been addressed and I don't have further comments. However, the aspects regarding my previous comments can be refined. In particular: 

Response #1 "Although it is true that we did not perform the statistical analysis in two separate groups, when performing the statistical calculations to relate the allelic variants with NAFLD (table 3), the model was initially adjusted for both HOMA-IR and BMI. Finally, taking into account that the statistical significance of the BMI (p= 0.008) was higher than HOMA-IR (p= 0.039) only the BMI was included, as they are physiologically related" First of all, model adjustement does not correct for effect modification (as in this case) rather only for confounding. So I suggest the Authors to perform a sub-analysis dividing patients for the presence of metabolic syndrome and/or insulin resistance. Otherwise, if Authors don't want to perform sub-analyses, I suggest otherwise to perform two different models of adjustement: one with HOMA-IR and one with BMI. Indeed, although the BMI shows higher significance than BMI at univariate analysis, significance values can be modified in the multivariate analysis.

Limitations of ultrasonography, as regards liver steatosis quantification, should be reported in the text (sorry, my previous comment was incomplete!)

Reviewer 2 Report

The Authors might be willing to address the following points:

-The Authors stated that they have considered five allelic variants that have been shown the greatest effect on triglyceride concentrations in patients. They should better justify the choice of these variants in both introduction and experimental plan.

Thus, the rationale of the study needs to be clarified. Still, it appears that the motivation of this study has been superficially justified.

- In the Abstract the Authors should better explain the sentence ‘We investigated the association between lipid and non-lipid risk factors and five allelic variants that have shown the greatest effect on triglyceride (TG) concentrations in patients with NAFLD’. What kind of traits the Authors considered?

The methods section is quite accurate; however, the statistics performed are not well addressed throughout the manuscript.

-The study design is poor, and the Authors should better elucidate how they stratify the groups according to the severity of the disease. In particular, why the Authors decided to put together patients with mild NAFLD and patients without NAFLD in Table 1?

-Table 2 Did the Authors checked the interaction between the two genotypes analyzed?

-The Authors should better explain why they have been chosen to measure the concentrations of lipoprotein (a). Indeed, the circulating levels of lipoprotein (a) has been previously reported to be downregulated in severe NAFLD with fibrosis. However, the debate is still opened, and many questions remain unsolved. Nowadays, lipoprotein (a) levels should not be considered as a marker of advanced liver damage.

-Since lipoprotein (a) levels have been previously correlated with diabetes and advanced fibrosis, did the Authors checked these associations in their cohort?

-Did the Authors checked the associations between lipoprotein (a) levels and the presence of each risk variant analyzed?

-Results: the overall analysis needs to be replicated and validated. The results are very poor and only elusive.

-Why the Authors decide to adjust the multivariate analysis only for gender, age, BMI, and not for genotypes? The Authors did not justify the choice of confounders and did not considered for example genetics as possible modifier.

-Why the Authors did not genotype their patients for PNPLA3 rs738409, TM6SF2 rs58542926, and MBOAT7 rs641738 variants, that are more closely associated with progressive liver damage?

-The Authors should better explain the statistical analyses performed in Table 3, elucidating dependent and independent variables and confounders.

-Table 3 need to be adjusted for genetic and enviromental modifiers.

Reviewer 3 Report

The authors fullfilled all my requests, so I am completely satisfied.

Author Response

Thank you for your suggestions